# Exploring Omega-3′s Impact on the Expression of Bone-Related Genes in Meagre (*Argyrosomus regius*)

**DOI:** 10.3390/biom14010056

**Published:** 2023-12-31

**Authors:** Leticia Luján-Amoraga, Belén Delgado-Martín, Cátia Lourenço-Marques, Paulo J. Gavaia, Jimena Bravo, Narcisa M. Bandarra, David Dominguez, Marisol S. Izquierdo, Pedro Pousão-Ferreira, Laura Ribeiro

**Affiliations:** 1Aquaculture Research Station (EPPO), Portuguese Institute for the Ocean and Atmosphere (IPMA), 8700-194 Olhão, Portugal; leticia.lujan@ipma.pt (L.L.-A.); catia.marques@ipma.pt (C.L.-M.); pedro.pousao@ipma.pt (P.P.-F.); 2Department of Microbiology and Crop Protection, Institute of Subtropical and Mediterranean Horticulture (IHSM-UMA-CSIC), 29010 Malaga, Spain; belendm@uma.es; 3Collaborative Laboratory on Sustainable and Smart Aquaculture (S2AQUACOLAB) Av. Parque Natural da Ria Formosa s/n, 8700-194 Olhão, Portugal; 4Centre of Marine Sciences (CCMAR), University of Algarve (UALG), 8005-139 Faro, Portugal; pgavaia@ualg.pt; 5Aquaculture Research Group (GIA), University of Las Palmas de Gran Canaria (ULPGC) Crta. Taliarte s/n, 35214 Telde, Spain; jimena.bravo@ulpgc.es (J.B.); david.dominguez@ulpgc.es (D.D.); marisol.izquierdo@ulpgc.es (M.S.I.); 6Division of Aquaculture, Upgrading, and Bioprospection (DivAV), Portuguese Institute for the Sea and Atmosphere (IPMA, IP), Rua Alfredo Magalhães Ramalho, 7, 1495-006 Lisbon, Portugal; narcisa@ipma.pt; 7CIIMAR, Interdisciplinary Centre of Marine and Environmental Research, University of Porto, Rua dos Bragas 289, 4050-123 Porto, Portugal

**Keywords:** fish larvae, skeletal development, differentially expressed genes, cartilage

## Abstract

Dietary supplementation with Omega-3 fatty acids seems to promote skeletal health. Therefore, their consumption at imbalanced or excessive levels has offered less beneficial or even prejudicial effects. Fish produced in aquaculture regimes are prone to develop abnormal skeletons. Although larval cultures are usually fed with diets supplemented with Omega-3 Long Chain Polyunsaturated fatty acids (LC-PUFAs), the lack of knowledge about the optimal requirements for fatty acids or about their impact on mechanisms that regulate skeletal development has impeded the design of diets that could improve bone formation during larval stages when the majority of skeletal anomalies appear. In this study, *Argyrosomus regius* larvae were fed different levels of Omega-3s (2.6% and 3.6% DW on diet) compared to a commercial diet. At 28 days after hatching (DAH), their transcriptomes were analyzed to study the modulation exerted in gene expression dynamics during larval development and identify impacted genes that can contribute to skeletal formation. Mainly, both levels of supplementation modulated bone-cell proliferation, the synthesis of bone components such as the extracellular matrix, and molecules involved in the interaction and signaling between bone components or in important cellular processes. The 2.6% level impacted several genes related to cartilage development, denoting a special impact on endochondral ossification, delaying this process. However, the 3.6% level seemed to accelerate this process by enhancing skeletal development. These results offered important insights into the impact of dietary Omega-3 LC-PUFAs on genes involved in the main molecular mechanism and cellular processes involved in skeletal development.

## 1. Introduction

The appearance of skeletal anomalies is an important problem for the aquaculture sector. However, the molecular mechanisms implicated in the onset of these malformations remain mostly unknown [1]. The skeletal system is a highly dynamic tissue continuously in formation and transformation through highly controlled bone modeling and remodeling processes involving different bone-cell types [2]. In advanced teleosts that present acellular bones, the main elements involved are involved mainly osteoblasts (OBs), responsible for the synthesis of new bone, and also osteoclasts (OCs), responsible for bone resorption [3]. It is well known that for correct bone formation, an equilibrium among the activity and function of these cells must be balanced. Despite the fact that the onset of these anomalies seems to be multifactorial, among the most important nutritional factors that control bone metabolism and modulate the development of the skeletal system are fatty acids, concretely the LC-PUFAs [4,5,6]. They are crucial in controlling several cellular and molecular pathways involved in bone modeling. According to the studies, these fatty acids seem to impact bone composition [7] through modulation of important molecular pathways controlling bone-cell differentiation [8] or the synthesis of some key bone components such as the extracellular matrix (ECM) [9]. Furthermore, as components of cell membranes, they can influence the fluidity and the behaviour of ECM components; membrane-bound enzymes and receptors or ion channels, affecting cellular signaling; the responsiveness to exogenous molecules; and the transport of proteins and minerals such as calcium (ion or mineral) [6,10]. In addition, they can impact bone formation directly or indirectly through their conversion into lipid mediators such as docosanoids and eicosanoids [11,12,13,14], which also present pro- or anti-osteogenic functions during skeletal formation (see reviews [12,13,15]). These nutrients and their mediators are also involved in key cellular processes such as calcium metabolism, oxidative stress, and inflammation that ultimately affect bone health [16]. However, despite the fact that the pathways by which these nutrients can control bone development have been mostly identified, the genes participating in these pathways that are most impacted by these nutritional factors have not been extensively studied and identified. LC-PUFAs are divided into two classes according to their structure: Omega-3 (n-3) and Omega-6 (n-6) [17]. The most biologically active among them are eicosapentaenoic acid (EPA) and docosahexaenoic acid (DHA) from the Omega-3 family and arachidonic acid (ARA) from the Omega-6 family. Generally, LC-PUFAs from the Omega-3 family are considered beneficial for bone health because they seem to increase bone mass and mineral content by stimulating the proliferation of cells involved in bone formation and inhibiting OC differentiation, such as decreasing the level of pro-inflammatory cytokines [18]. Still, excessive levels seem to increase oxidative stress, ultimately affecting bone health [19]. Regarding ARA, the current literature denotes that high levels negatively affect skeletal development because it can increase bone resorption and reduce bone formation [20,21]. These effects can be related to the ability to increase the production of pro-inflammatory cytokines or ROS [22]. Due to this wide complexity, information about the impact of the different PUFAs on molecular mechanisms and cellular pathways controlling skeletal development is still scarce, mainly in some groups of organisms, as in common teleosts employed in the aquaculture sector.

However, since aquaculture-reared fish usually present a high number of skeletal malformations, producers have shown increasing interest in the potential of these nutrients to improve skeletal health and reduce the economic losses associated with deformed individuals. Since most malformations occurring in aquaculture production originate during larval stages and these cultures are routinely fed with formulations containing significant levels of these nutrients, it is believed that the optimization of dietary levels during this period can contribute to reducing the occurrence of skeleton deformities in farmed fish and improve production rates [9]. Despite the fact that studies have demonstrated the beneficial effect of adequate Omega-3 supplementation in larval culture performance [23,24,25], only a few have shown that these nutrients are able to improve skeletal formation and reduce the incidence of some types of bone anomalies [26,27]. However, the scarcity of comparable results has impeded the design of diets that could improve skeletal formation during these life stages and the acquisition of knowledge about the molecular mechanisms involved in the control of skeletal formation carried out by PUFAs.

Currently, thanks to the advances in genetic sequencing technologies and computational methods, we can use genomic and transcriptomic approaches to gain information to improve fish health and quality in the aquaculture sector [28]. In fact, it is believed that gaining knowledge about genes that are involved in the molecular pathways employed by these nutrients is a promising tool for the aquaculture industry with which to reduce the prevalence of deformities [29]. The analysis of the differential expression of genes encoding important bone biomarkers, enzymes, cytokines, or growth factors controlling bone metabolism can be an effective method to denote the impact of dietary LC-PUFAs during skeletal formation. Studies evaluating alterations in the genetic expression of skeletal-related specific genes exerted by dietary PUFAs are limited in fish [30,31,32]. Meagre (*Argyrosomus regius)* is a promising candidate for the diversification of Mediterranean aquaculture, based on its fast growth, high flesh quality, and economic value [33]. Their fast development rates and easy adaptation to captivity have contributed to a fourfold increase in meagre production, between 2010 and 2019 (EUMOFA, 2022), reaching around 55,000 tons in the European Union in 2019. The research developed for this species regarding broodstock management or optimization of rearing protocols [34], nutritional aspects, and feed optimizations [35,36,37], among others, have contributed to the success of meagre production. Still, the molecular mechanism involved in their skeletal development and the impact of functional diets in this process is scarce. The knowledge about these mechanisms can help to optimize maintaining their skeletal health. Hence, a transcriptomic study was performed to evaluate the impact of supplementation of different levels of Omega-3 on the gene expression profile during the bone formation of meagre. The wide quantity of information obtained in this study about the genetic modulation exerted by these nutrients during skeletal formation can be useful to design diets that can improve skeletal development maintaining bone health and minimizing the incidence of skeletal deformities during aquaculture production.

## 2. Materials and Methods

### 2.1. Larval Culture

Eggs were obtained from a natural posture of F1 breeders maintained in captivity at the installations of the Estação Piloto de Piscicultura de Olhão (EPPO) belonging to the Instituto Português do Mar e da Atmosfera (IPMA), where the experiment was carried out. Eggs were incubated in 200 L fiberglass tanks until larvae hatched. Once hatched, a total of 10,000 larvae were distributed in each tank, establishing an initial density of 33 lv/L. The selected temperature was 21–22 °C. Dissolved oxygen was maintained at over 90%, and the photoperiod selected was 14 h of light and 10 h of darkness. The tanks were provided with air-lift systems for aeration and were supplied with pre-filtered, UV-sterilized seawater.

### 2.2. Feeding Protocol

Triplicate groups of *Argyrososmus regius* larvae were settled to test different experimental diets. The diets were formulated and produced by SPAROS Lda (Olhão, Portugal). The formulations for Medium Diet (MD) and High Diet (HD) presented increasing levels of LC-PUFAs Omega-3: 2.6 and 3.6 (% DW), respectively. The ingredients of the experimental diets are listed in Appendix A. Furthermore, a Reference Diet (RD), to compare the efficacy of our experimental diets in culture performance, was employed using a commercial diet from SPAROS Lda (Olhão, Portugal) that presented a similar proportion of proteins and lipids as the experimental diets. The DHA/EPA ratios for the different diets, RD, MD, and HD, were 0.97, 0.71, and 1.42, respectively.

### 2.3. Sampling

The number of fish sampled for all experimental procedures was estimated according to the minimum number of animals necessary to provide reliable and robust statistical results. At 28 days after hatching (DAH), six larvae from each triplicate from each treatment were collected, rinsed with distilled water, immediately transferred into tubes containing RNAlater (Qiagen, Hilden, Germany), placed at 4 °C for 24 h, and then stored at −20 °C until use for genetic purposes. At this age, meagre larvae have formed almost all the elements of the vertebral column, whereas cartilage differentiation and ECM synthesis associated with endochondral ossification and the mineralization of the axial skeleton are still ongoing [38].

### 2.4. RNA Extraction

The RNA was extracted at 28 DAH via disruption of the samples in liquid nitrogen using a mortar and pestle. RNA was extracted using the RNA extraction Kit NZYtotal RNA isolation kit (NZYtech, Lisbon, Portugal), according to the manufacturer’s instructions. RNA yield and purity were determined by measuring the absorbance at 260 and 280 nm using a NanoDrop DS-11FX spectrophotometer (DeNovix, Wilmington, Delaware, USA), and the integrity was assessed using 1.5% agarose gel electrophoresis. The results were read using ChemiDoc XRS+ (Bio-Rad, Hercules, California, USA) with Image Lab software.

### 2.5. Sequencing

Extracted RNA was sent to Eurofins Genomics Europe Sequencing GmbH (Kostanz, Germany) and employed for library construction. Prepared cDNA libraries were sequenced with Illumina. Paired-end reads of 150 base pair (bp) lengths were generated per sample. Library type: strand-specific cDNA library. Library preparation methods: purification of poly-A-containing mRNA molecules, mRNA fragmentation, random primed cDNA synthesis (strand-specific), adapter ligation, and adapter-specific PCR amplification Illumina sequencing. Sequencing method: Illumina paired-end, random primed cDNA synthesis (strand-specific).

### 2.6. RNA-Seq Data Processing

Raw reads were processed using a bioinformatic pipeline, as indicated in Figure 1. To obtain high-quality clean reads, the raw reads were pre-processed with SeqTrimBB (v2.1.8), and mapping was performed with BWA-MEM2 (v2.2.1). To identify differentially expressed genes (DEGs) across samples or groups, DEgenesHunter (v1.0) software was employed. Genes were considered as differentially expressed (DEG) if the [fold-change] > 1.5 and the adjusted *p*-value < 0.05 (calculated with the false discovery rate (FDR)). Three comparisons between dietary treatments were carried out: ‘Medium Diet’ vs. ‘Reference Diet’ (MD vs. RD), ‘High Diet’ vs. ‘Reference Diet’ (HD vs. RD), and ‘High Diet’ vs. ‘Medium Diet’ (HD vs. MD).

### 2.7. Functional Enrichment Analysis

Enrichment analysis is a method that employs data mining to identify whether genes from pre-defined sets/are present more than expected in the dataset. This analysis permits elucidation of the functional roles of DEGs during larval development. The functional enrichment of GO terms and KEGG pathway analysis was performed with in-house R scripts (using the computational resources of the Andalusian Bioinformatics Platform, located at the University of Málaga in Spain). Since no available reference exist for *Argyrosomus regius*, the reads were mapped to the *Larimichthys crocea* reference genome (RefSeq assembly accession: GCF_000972845.2) that was annotated with eggNOG (v2.1.6). According to Phylogenomic analysis [39], both species are quite close and present a high homology [40]. These results support the mapping of the meagre genome employing this *Larimichthys crocea* as reference, contributing improvement in the scarce information available for meagre.

To find out the most significant processes in which the genes present in our data are involved according to enrichment analysis, all DEGs were mapped to the available terms in the GO and KEEG databases. Firstly, a GSEA (Gene Set Enrichment Analysis) was performed employing as input all the genes expressed in the transcriptomic analysis, even those that did not pass the double thresholds (*p*-value < 0.05 (cut-off at 5% FDR) and a fold-change of either ≥1.5 or ≤1.5) to be considered as DEGs. All expressed genes were mapped to the KEGG database (http://www.genome.jp/kegg/, accessed on 11 October 2022), and significantly enriched KEGG pathways were identified in order to output a general overview of all modulated pathways due to the effect of the dietary treatment across the large network of expressed genes. Then, the pathways that were significantly enriched (*p*-value < 0.05) were further analyzed. Nevertheless, this approach has certain limitations in obtaining information about the true impact of PUFAs on individual DEGs. Therefore, to improve the results and increase the specificity at the individual gene level, an ORA (Over-Representation Analysis) was performed employing only genes that passed the double thresholds (*p*-value < 0.05 (cut-off at 5% FDR and a fold-change of either ≥1.5 or ≤1.5). Furthermore, through an ORA analysis, the GO terms in which the DEGs obtained in the different comparisons were overexpressed were retrieved. GO terms were classified into three subgroups, namely biological process (BP), cellular component (CC), and molecular function (MF).

### 2.8. Clustering

Gene expression clustering provides a general and organized overview of the data without becoming lost among the thousands of individual genes [41]. In addition, this analysis permits a deeper understanding of the modulatory effect of LC-PUFAs in individual genes of interest and investigates their function as it is believed that the genes involved in the same function can be grouped together because they present a similar regulation of their genic expression. To this end, a list of interesting genes obtained as DEGs for some of the comparisons (HD vs. RD, MD vs. RD, and HD vs. MD) and that participated in the most overrepresented pathways obtained in the GO and KEEG enrichment analysis were selected for further study. According to the available bibliography, these genes are involved in skeletal development and bone metabolism in vertebrates, and in some cases, imbalances in their expression have been directly related to alterations of bone formation. Hierarchical cluster analysis was performed using Euclidean distances to compare the similarity in the expression between genes. Furthermore, Ward’s method was employed to join groups of genes and generate a dendrogram. Cluster analysis is used to identify similar variations instead of significant changes. It allows us to group together genes that are highly correlated in their expression.

### 2.9. Quantitative Real-Time Polymerase Chain Reaction (qRT-PCR) for Validation

To validate the RNA sequencing and transcriptome analysis, the expression of a set of bone-related genes was evaluated using qRT-PCR. The genes selected were *sp7* (osterix), *mgp* (matrix gla protein), and *bmp2* (bone morphogenetic protein 2), which are identified in Table 1. Total RNA was extracted and quantified as described in the previous Section 2.4. The cDNA was generated from 1 µg of total RNA from the larvae with 28 DAH using the M-MuLV first-strand cDNA synthesis Kit (NZYtech, Lisbon, Portugal) following the manufacturer’s protocol. Synthesis of cDNA comprises two main steps: elimination of genomic DNA and reverse transcription into cDNA, which was used for PCR quantification of the specific genes. The cDNA was stored at −20 °C until further analysis. Quantitative analysis of genes considered bone remodeling biomarkers was undertaken, and their expression was studied using quantitative real-time polymerase chain reaction (qRT-PCR). To design the primer of interest listed in Table 1, the nucleotide sequences of these genes from this species were blasted using the NCBI database. Then, based on the FASTA, primers were designed on the identified conserved regions (Tm 60 °C; G/C 45–55%) using Primer3plus (www.primer3plus.com/, accessed on 21 March 2022). To determine primer efficiency, five serial two-fold dilutions of cDNA mix of all samples were prepared, and efficiency was calculated from the slope of the regression line of the quantification cycle (Ct) versus the log 10 of the different cDNA solutions. Quantitative PCR using SensiFAST master mix was applied to determine the relative expression of selected genes using the specific primer sets for each gene. The qPCR reactions were set up in a final 20 µL volume with 6.4 µL of ultrapure water (NZYtech, Lisboa, Portugal), 10 µL of SensiFAST, 2 µL of total cDNA, and 400 nM of both forward and reverse primers. The PCR amplification was started with an initial polymerase activation for 10 min at 95 °C, and then amplification via 40 cycles of PCR were as follows: denaturation at 95 °C for 15 s and annealing and extension at 60 °C for 30 s (Bio-Rad CFX Connect–Real-time system). Each sample was run in duplicate, and the specificity of the reaction was verified via melting curve analysis. Data were normalized to *Ef1* and *Tub* (because of its abundance and Ct value consistency among treatments) using the 2^−ΔΔCt^ method.

## 3. Results

### 3.1. Transcriptome Assembly and Annotation

The transcriptomes of whole larvae at 28 DAH were analyzed. Nine cDNA samples (three replicates per treatment) were sequenced, and more than 150 million bp paired-end reads were generated for each library. Reads were processed for subsequent transcriptome analysis (Table 2). The reads were mapped to the *Larimichthys crocea* genome (RefSeq assembly accession: GCF_000972845.2), obtaining average mapping rates of 67.60%, 67.39%, and 69.21% for Reference Diet (RD), Medium Diet (MD), and High Diet (HD), respectively. More than 13,500 genes were assigned.

### 3.2. Differentially Expressed Genes (DEGs)

To identify DEGs in the *Argyrosomus regius* larvae during development, pair-wise comparisons were performed among the different dietary treatments. The impact on total transcriptome expression and the number of DEGs obtained from the different comparisons can be observed in the MA plots (Figure 2A). A high proportion of DEGs was found when the Medium Diet and Reference Diet were compared (648 up- and 441 down-regulated, Figure 2(Aa)) and when the High Diet and Medium Diet were compared (345 up- and 618 down-regulated, Figure 2(Ac)). However, a significantly lower number of DEGs (50 up- and 30 down-regulated Figure 2(Ab)) was found after a comparison between the HD and MD. Using Venn diagram analysis (Figure 2B), it was observed that there were only three DEGs overlapped among the three comparisons between diets. Furthermore, 457 DEGs overlapped between the comparisons MD vs. RD and HD vs. MD, 5 overlapped between the comparisons HD vs. RD and HD vs. MD, and 47 DEGs overlapped between the comparisons MD vs. RD and HD vs. RD. Furthermore, 582 DEGs were identified uniquely in MD vs. RD, 498 DEGs were identified uniquely in HD vs. MD, and 25 were identified uniquely in HD vs. RD.

### 3.3. Detailed Transcriptomic Data Analysis Based on KEGG Enrichment

In Figure 3 are shown the GSEA results obtained from the different comparisons. The dot-plots present the most enriched pathways for each comparison considering all the genes expressed in the dataset (MD vs. RD, HD vs. RD, and HD vs. MD in Figure 3A, Figure 3B, and Figure 3C, respectively). These results were divided into two categories regarding whether the total expression calculated for each pathway was activated or suppressed, considering the relative expression of all genes included in each term. Therefore, it was observed that the levels of supplementation with Omega-3 in the diet significantly enriched (*p*.adj < 0.05) different pathways during larval development and metabolism. Mainly, the MD in comparison to the RD (Figure 3A) activated processes related to drug metabolism-cytochrome P450 (ko00982), metabolism of xenobiotics by cytochrome P450 (ko00980), oxidative phosphorylation (ko00190), glutathione metabolism (ko00480), or the VEGF signaling pathway (ko04370). Some of these cited pathways presented high gene ratios and an elevated number of genes correlated. Contrarily, the Wnt signaling pathway (ko04310), calcium signaling pathway (ko04020), and FoxO signaling pathway (ko04068) were among the most enriched pathways identified from the suppressed terms, and they presented notably different gene ratios.

In the facet plot B (Figure 3B), the pathways most affected by the HD in comparison to RD are identified. Among those activated that were most enriched, pathways were found that were related to communication and binding among cells and with the ECM, such as ECM-receptor interaction (ko04512), regulation of actin cytoskeleton (ko04810), and focal adhesion (ko04510). On the contrary, several suppressed genes were correlated to terms involved in the biosynthesis of lipids, as in steroid biosynthesis (ko00100), in the biosynthesis of unsaturated fatty acids (ko01040) and their elongation (ko00062), or even in the adipocytokine signaling pathway (ko04920). In the facet dot-plot C (Figure 3C), it can be observed that an elevated number of pathways were highly enriched in both cases among the suppressed and activated terms when both levels of supplementation were compared (MD and HD). Furthermore, a wide number of genes were related to each term according to the GSEA results.

However, to obtain more precise information about the role of genes that were more impacted, an ORA analysis was performed employing only DEGs. Mainly, the larvae feed with the MD in comparison to the RD (Figure 3D) enriched pathways related to ribosome (ko03010), ECM-receptor interaction (ko04512), arachidonic acid metabolism (ko00590), metabolism of xenobiotics by cytochrome p450 (ko00980), or drug metabolism-cytochrome p450 (ko0098). Among the genes involved in ECM-receptor interaction (ko04512), several were observed that were related to the synthesis of collagenous and other non-collagenous proteins such as integrins, tenascins, nidogens, or cadherins that exert a crucial role in the binding and organization of the different ECM components. Furthermore, this diet also impacted genes involved in the synthesis of lipid mediators such as leukotrienes or prostaglandins that were included in the pathways: arachidonic acid metabolism (ko00590), metabolism of xenobiotics by cytochrome p450 (ko00980), and drug metabolism-cytochrome p450 (ko00982). Furthermore, the DEGS in this comparison also enriched pathways involved in the production and detoxification of toxic compounds (ko00480) derived from the metabolism of fatty acids as alcohol dehydrogenases, glucuronosyltransferase, or glutathione peroxidases. Some of the DEGs involved in these pathways were also present in other less significantly enriched pathways, as can be observed in the upset-plot (Figure 3G). For instance, processes such as LA-metabolism (ko00591) and ALA-metabolism (ko00592) shared genes such as phospholipase A2 (*PLA2)* with ARA metabolism (ko00590). A similar situation occurs among glutathione metabolism (ko00480), ARA metabolism (ko00590), or metabolic pathways involved in the metabolism of drugs (ko00982) and xenobiotics (ko00980), with glutathione S-transferase (*GST)* participating in all of them. In the same way, but with a low number of shared genes, it can be observed that the pathway metabolism of glutathione (ko00480) presented genes that participate in the metabolism of ARA (ko00590) as some glutathione peroxidases or also with the pathway’s ECM-receptor interaction (ko04512) and PPAR signaling (ko03320) sharing the gene coding for the molecule CD36.

The HD in comparison to the RD (Figure 3E) enriched mainly the pathways involved in focal adhesion (ko04510) and ECM-receptor interaction (ko04512) that moreover presented the highest number of genes affected and the highest gene ratios among the identified pathways. These pathways also shared several genes coding for collagenous and non-collagenous proteins that usually are involved in other crucial cellular functions. Nevertheless, different terms involved in pathways associated to PUFA metabolism were also enriched. Concretely, ARA (ko00590), LA (ko00591), and ALA (ko00592) metabolism that share some DEGs as can be observed in the upset-plot (Figure 3H). Furthermore, some processes were differently regulated by the two experimental diets MD and HD. In Figure 3E, pathways differentially impacted by the two different levels were identified. For instance, enriched pathways were associated to Ribosome (ko03010), PPARs signaling (ko03320), or steroid biosynthesis (ko00100) that did not present shared affected genes among them or with other KEEG terms identified (Figure 3I). Moreover, they also regulated drug metabolism (ko00982), metabolism of xenobiotics by cytochrome p450 (ko00980), or drug metabolism-other enzymes (ko00983) differently. However, in this case these pathways shared some genes among them including the glutathione metabolism pathway (ko00480). Through the results obtained, the enriched pathways that were differently impacted between treatments and that are known to be involved in the control carried out by PUFAs in bone metabolism were selected for further investigation.

### 3.4. Detailed Transcriptomic Data Analysis Based on GO Terms Enrichment

In the comparison (MD vs. RD) the up-regulated DEGs correlated to GO categories related to CC presented the highest number of DEGs associated to the extracellular region (GO:0005576), with 18% of the total of DEGs participating, and extracellular space with 9% of DEGs significantly correlated. However, regarding the BP categories, the pathway presenting a higher number of associated genes was a small-molecule metabolic process (GO:0044281) with 17% of the DEGs up-regulated. Furthermore, in the MF category, 2% of DEGs participated in fatty acid binding (GO:0005504). However, repressed genes related to CC were also mostly associated with the membrane (GO:0016020), presenting this term as 52% of the DEGs correlated. Concretely, 42% of them associated to the cell periphery (GO:0071944), 39% associated to the plasma membrane (GO:0005886), 36% associated with intrinsic components of the plasma membrane (GO:0031226), and 34% to integral components of the plasma membrane (GO:0005887). In the case of down-regulated genes associated with BP, terms related to cell communication (GO:0007154) were obtained with 47% of DEGs involved, signaling (GO:0023052) with 46%, system process (GO:0003008) with 23%, ion transport (GO:0006811) with 16%, or cell death (GO:0008219) with a 23% of DEGs participating in this process. However, a lower number of down-regulated DEGs were annotated in GO terms related to MF. In this class, most of the enriched terms were related to transmembrane transporters’ activity, including the transporter for carboxylic acids (GO:0046943), organic anions (GO:0008514), and organic acids (GO:0005342), with around 4%-5% of DEGs participating in each of them. On the other side, when comparing the diet with the highest level of Omega-3 (HD) and the reference diet (RD), several up-regulated genes were related to CC terms: 40% of DEGs were associated with extracellular region (GO:0005576), while 18% were correlated with supramolecular polymers (GO:0099081) or a 12% with contractile fibres (GO:0043292). In fact, several enriched pathways were related to collagen, as for instance the 10% of DEGs that were involved in collagen trimer (GO:0005581). Regarding the terms associated with BP, 16% of DEGs participated in extracellular matrix organization (GO:0030198), while 16% were included in extracellular structure organization (GO:0043062), and another 16% were involved with external encapsulating structure organization (GO:0045229). Furthermore, important biological processes related to cytoskeleton and filament sliding were enriched, as actin-mediated cell contraction (GO:0070252) with 10% of identified DEGs correlated. Again, attending to the DEGs associated with MF, the results indicated that 12% of DEGS were involved in the extracellular matrix structural constituent (GO:0005201), 6% in the extracellular matrix structural constituent conferring tensile strength (GO:0030020), and 20% in structural molecule activity (GO:0005198). On the contrary, attending to DEGs down-regulated in this comparison (HD vs. RD) and participating in the categories of CC, most of them were related to organellar membranes: 6% were coded for the extrinsic component of the organelle membrane (GO:0031312). However, regarding the BP terms, 31% were associated with the response to lipids (GO:0033993), 28% with the cellular response to cytokine stimulus (GO:0071345), and 12% were associated with positive regulation of fat cell differentiation (GO:0045600).

Again, when comparing both experimental diets, what was appreciated was that both diets impacted the pattern of expression of several genes during development differently. This different regulation was reflected in the elevated number of enriched terms and the higher number of DEGS identified in response to the different levels of PUFA Omega-3s in both diets. Several up-regulated genes were grouped in the CC categories as involved in the cell periphery (GO:0071944), cell junction (GO:0030054), and plasma membrane region (GO:0098590). Furthermore, enriched terms related to supramolecular complexes (GO:0099080) as related to contractile fibres (GO:0043292) or the collagen-containing extracellular matrix (GO:0062023) were obtained. Furthermore, regarding the DEGS that were related to BP, several terms were found involved in larval morphogenesis and development, and 12% were involved in muscle organ development (GO:0007517) or eye development (GO:0042460).

Also observed were several genes coding for proteins involved in cell adhesion (GO:0007155) and cell morphogenesis involved in differentiation (GO:0000904), 28% in the movement of cells or subcellular components (GO:0006928) and 22% in cellular component morphogenesis (GO:0032989) or in anatomical structure morphogenesis (GO:0009653) and animal organ development (GO:0048513). Finally, among the pathways significantly enriched regarding the MF, several functions related to musculoskeletal development were also found, as in actin filament binding (GO:0051015), a 4% structural constituent of muscle (GO:0008307), or a 2% structural molecule activity conferring elasticity (GO:0097493). Some of the most enriched pathways were also identified with a high number of genes associated, such as calcium ion binding (GO:0005509) and protein binding (GO:0005515). Regarding down-regulated genes, despite the fact that the terms in the CC and BP categories were mostly related to ribosome or ribosomal processes, among the MF categories pathways were included that were related to the metabolism of fatty acids as occurs with 2% of DEGs involved in fatty acid binding (GO:0005504) and the 7% that were associated with oxidoreductase activity (GO:0016491) or with the metabolism of their products or subproducts as for instance, the 2% of DEGS that were involved in antioxidant activity (GO:0016209).

From the enrichment analyses, it was confirmed that Omega-3 levels may coordinate the regulation of skeletal development by modulating the expression of target genes involved in several pathways.

### 3.5. Screening and Clustering of Interesting DEGs Retrieved among KEGG and GO Pathways

Among the genes impacted in relevant pathways that modulate skeletal development under the control of dietary PUFAs, a list of interesting genes that exert important roles in bone metabolism was selected to better understand the modulatory effect of these nutrients in skeletal development. Hence, the list of 96 genes, obtained as DEGs for some of the comparisons (HD vs. RD, MD vs. RD, and HD vs. MD) was divided into three panels regarding if they present a major role in the synthesis, organization, and binding of the extracellular matrix; the differentiation of skeletal cells (osteoblast, osteoclast, and chondrocytes); or in the modulation of different cellular processes known to be affected by the dietary fatty acids such as calcium metabolism, oxidative stress, or in the synthesis of fatty acids and their derived mediators among other inflammatory molecules. The figures show the hierarchical clustering of the selected genes.

In the first panel (Figure 4A,B), genes selected among KEEGS and GO pathways were grouped together because of the code for coupling factors, extracellular-related proteins or signaling molecules released from the matrix or produced within the bone-cell lineage, and other cell types that regulate important transcriptional mechanisms controlling bone-cell differentiation. DEGS grouped in this panel were clustered in two main groups according to their differences in Euclidean distances. The first group presented most of the genes that were down-regulated in the MD in comparison to the RD. Among the genes involved in this cluster, several corresponded to transcriptional factors such as *alx4*, *ppargc1a*, *sox10*, *znf219*, *mef2d*, and *mef2c* or *asxl2*. However, none of these genes showed an expression significantly different from HD to that of RD. Nevertheless, *bcl11b* was significantly up-regulated in HD in comparison to MD as also occurred with *sox10*. Critical signaling factors were also been included, such as growth factors or cytokines produced by different cell types that are crucial regulators of molecular pathways modulating osteogenesis (BMPs, NOTCH, WNTs). Among them, receptors such as *lrp6*, *ddr2l*, or notch receptors such as *notch2* and *notch3* were found, and also genes coding for their ligands as *jag2*. Despite some of these cited molecules passing the FC threshold, they did not pass the quality control measured with the FDR value in order to be considered true DEGs in the comparison between MD and RD, and only *jag2* was considered a true DEG in this comparison. Contrarily, the genes *lrp6*, *notch2*, and *notch3* passed both thresholds and are considered as true up-regulated DEGs in the comparison between both experimental diets, denoting a different impact in their expression related to the levels of Omega-3 supplementation. Other signaling molecules, such as the cytokine *tgfβ2*, were down-regulated in the MD.

Furthermore, were also selected genes that, despite also participating in ECM functions such as binding, adhesions, and interactions among cytoskeleton, ECM, and other cellular components, are expressed by bone cells during the different stages of maturation and exert an important role in signaling and communication processes that regulate cell phenotype and differentiation. Some of them were down-regulated in the MD regarding the RD as membrane proteoglycans (*sdc-2*) and membranal (*rflna* and *lama3*) or binding proteins (*fat4* and *cdh-2*). However, different regulation between both levels of supplementation was observed in some genes involved in these processes. The HD increased expression of adhesion molecules as *mxra8* and genes more related to interactions with the cytoskeleton such as *macf1a* and *pkd1a*. The second main cluster was divided in two main groups of genes at the same time. One of these groups was formed by genes that were up-regulated in the MD in comparison to the RD, but they did not present significant differences in the HD in comparison to the RD. This subgroup was mainly formed by genes coding for proteins involved in bone-cell interactions with regulatory molecules as members of the Wnt pathway and controlling the expression of osteogenic markers, as for instance *sdc-3*, *mustn1a*, and *tmem119*. Furthermore, *cldn-18*, *ctrh1*, and *pkdcc* were included in this cluster, and they also seem to impact the expression of bone-developing markers, but the molecular mechanism by which they exert this effect has been less clarified. Two genes that were not related to ECM interactions were included in this group. They were the transcription factors *foxa2* and *ptch3d* (a receptor that is involved in Ihh pathway). The second group, however, involved genes that were up-regulated in both the MD and the HD experimental treatments compared to the RD. Only two genes formed this cluster *snorc* (a proteoglycan of chondrocytes-secreted ECM) and the growth factor *wisp3* (also named *ccn6*, which is involved in the Wnt pathway).

Figure 4C,D show the clustering of genes involved in ECM synthesis and maintenance. When observing the aggrupation of genes in this panel according to their expression, two main clusters can be observed. The first main cluster is divided into two groups. The first group corresponds with genes that were up-regulated in the larvae from MD in comparison with the RD, but their expression does not change significantly in the larvae from the HD. Mainly, this group was formed by genes considered markers of mature cells that synthesize the ECM as mature osteoblasts and chondrocytes and for components of the ECM. Among them, genes coding for the main ECM components of cartilage and markers of proliferating chondrocytes such as *col2a1* variant B were observed, but they also included genes coding for non-collagenous proteins involved in mineralization such as *mgp* and *alp3*, or proteins involved in the metabolism of members of the proteoglycan family, such as *hapln1* or *chst3a*. Any of these genes showed differences in their regulation between the different levels of Omega-3 in MD and HD.

The second group presented genes that were significantly down-regulated in the MD but were not significantly affected in the fish treated with the HD in comparison with the RD. In this group, genes coding for components of the extracellular matrix of OB as for the proteoglycan Keratocan (*kera*) and genes coding for collagen proteins as *col13a1* and *col6a3* (usually present in basement membranes) were included, while *col6a4l* was also included but was differently regulated by the two levels of Omega-3, being significantly up-regulated in the larvae fed with the HD. Furthermore, in this cluster, several non-collagenous proteins that allow the cell adhesion, organization, and binding of the different cell components and the ECM but that also are involved in other important processes as *sema3* (involved in signaling between bone cells) were observed. Others such as *nid1*, *thbs1*, and *angpt2*, which are important factors involved in membrane organization and vascularization of mineralized tissues were also present. However, most genes coding for non-collagenous proteins were differently regulated among the two experimental diets with supplementation of Omega-3 as *nid1*, *nid2*, *thbs2*, *angp1*, *cilp2*, and *α11β1* (integrin that is expressed in osteoblast). All these genes were up-regulated in the HD in comparison to the MD, whereas among the genes present in this group, only *mmp13*, coding for a matrix metalloprotease involved in degradation and resorption of the bone matrix, was down-regulated in the HD compared to the RD. The last cluster presented genes that were significantly up-regulated in both experimental treatments in comparison with the RD, but in a more pronounced way in the moderate level of Omega-3 in the MD. This cluster mostly entailed proteins involved in cartilage development and in the synthesis of ECM components. Among them, collagens present in the cartilaginous matrix (*col2a1*, *col9a1*, *col9a2*, and *col9a3*), proteoglycans (*acanb* or *acanb-like*), or proteins involved in their synthesis were identified. In addition, structural proteins that mediate signaling interactions and binding between matrix constituents were included in this group, as *cnmd1*, *matn1*, and *tnxba*. All the genes in this cluster passed the threshold of 1.5 for fold-change to be considered differentially up-regulated in both treatments with Omega-3 supplementation. However, some did not accomplish the FDR value in order to be considered DEGs in the treatment with the highest level of Omega-3 supplementation (HD), and they were all more related to the cartilage ECM (*acanb*, *acanb-like*, *col2a1a*, and *csgalnact1*).

The third panel included genes involved in important cellular pathways under the control of PUFAs that have been probed to regulate bone metabolism (Figure 4E,F). Hence, genes involved in calcium metabolism and signaling, synthesis and transport of fatty acids and their derived lipid mediators, and in inflammatory processes were selected. In this panel, two main clusters were observed. The first cluster is again divided into two groups. The first group included genes that were up-regulated in the MD in comparison with the RD. Among these genes, several genes were found to be involved in the antioxidant system. Several glutathione peroxidases (*gpx1*, *gpx4a*, and *gpx8*) and other antioxidant proteins such as *gsto1* or perredoxins such as *prdx1*, *prdx2a*, and *prx2b*, were identified. Furthermore, some DEGs coding for proteins involved in calcium metabolism and mineralization such as *anxa2* or s100 proteins such as *s100p* and *s100a10a* were identified. Furthermore, some genes involved in inflammatory processes such as *ephx2*, coding for a hydrolase, which converts epoxy fatty acids (EpFAs) produced by cytochrome P450 enzymes, were also up-regulated. The second group of this cluster was mainly composed of genes that were highly up-regulated in the MD, while none of them was significantly affected in the HD with regard to the RD. This group included genes involved in the transport of fatty acids (*fabp2*, *fabp6*, or *CD36* molecule) or calcium (*rcan1*, *s100a1*, and the hypocalcemic factor *sst*). The second main cluster, however, included down-regulated genes in both experimental treatments in comparison to the RD, but in a more pronounced way in the MD. Among them, genes coding for calcium-dependent proteins involved in cellular signaling were included, such as *camk2a* and *camk2b1*, as well as genes involved in the synthesis of lipid mediators related to inflammatory processes, such as leukotriene B4 (*ltb4*) or others involved in the synthesis and metabolism of prostaglandins such as *ptgis*, *ptgs2b*, or *ptger2a*. Finally, this cluster also included *pparα*, which is involved in several functions related to fatty acid metabolism, such as oxidation.

## 4. Discussion

This study describes for the first time the gene expression pattern of the skeletal formation of *Argyrosomus regius* larvae, fed diets with different levels of Omega-3 supplementation. The results of this study revealed important insights into the molecular basis of bone development and the role of PUFAs in modulating this process, by identifying the impact of these formulations on individual genes participating in important cellular and molecular pathways controlling bone metabolism.

Although macroscopically the skeleton seems to be a static organ, it is one of the most metabolically active tissues at the microscopic level. It is constantly changing and is responsible for the growth and maintenance of body shape and skeletal health through two complex mechanisms: modeling and remodeling [43]. In these processes, different bone-cell types are involved in advanced teleosts with acellular bone [44]: chondrocytes (CHs) and osteoblasts (OBs), which are mesenchymal-derived cells responsible for bone formation and extracellular matrix deposition, and osteoclasts (OCs), with hematopoietic origin, are responsible for bone resorption [45]. Furthermore, two main patterns of bone formation have been identified in this species. On one hand, intramembranous ossification starts with the condensation of mesenchymal cells that directly differentiate into osteoblasts without a cartilaginous phase, as occurs in the vertebral bodies and intermuscular bones [46]. However, endochondral ossification is characterized by cells from the mesenchyme that coalesce, condense, and differentiate into chondrocytes that form cartilage, which is then eventually replaced by bone [47]. This process is usually accompanied by perichondral ossification. In this process, the bone starts to be produced around the periphery of the cartilage [48]. In ossification that employs a cartilage template, chondroblasts change their genetic program and maturate to CHs that become hypertrophic and die [49]. This is followed by the resorption and calcification of the cartilage extracellular matrix that they have constructed, mainly formed by type II collagen and aggrecan. During this stage, the matrix is invaded and replaced by the cells that will become the constituent cells of bones, such as blood vessels, bone marrow cells, osteoclasts for the removal of this cartilage matrix, and osteoblasts that deposit bone on the remnants of the cartilage matrix [50].

Bone is continuously remodeled throughout life and is regulated at several levels to maintain the equilibrium between functional osteoblasts and osteoclasts [51]. The number and activity of bone cells are further controlled by transcriptional and epigenetic mechanisms that are also highly regulated by hormonal and local regulatory proteins such as growth or transcriptional factors or cytokines [52,53,54]. Furthermore, in recent years the control performed by mechanical strain, cell–cell interactions, and cell–matrix interactions [55,56]. Due to the complexity of all these regulatory mechanisms and molecules involved, imbalances or alterations in these factors can result in incorrect bone formation and lead to the onset of skeletal anomalies [57]. Skeletal deformities in fish entail abnormal transformations of normal bone and cartilage structures throughout metaplastic changes, trans-differentiation, and the development of intermediate tissues, among other pathological events that lead to the appearance of a complex mixture of skeletal alterations that differ from their normal morphology [45,58,59,60,61,62,63,64,65,66,67]. Previous studies have indicated that bone abnormalities originate mainly during early developmental stages when the skeleton is being formed (i.e., during chondrogenic and early osteogenic differentiation periods) [68,69,70]. Due to these factors, 28DAH larvae were used, since these processes were still not completed.

Previous studies have indicated that the onset of skeletal disorders is multifactorial, linked to alterations in biotic and abiotic factors that can regulate the systemic and local factors controlling bone metabolism [1,71,72,73,74]. Among the nutrients that can control skeletal development, dietary highly polyunsaturated fatty acids seem to play a crucial role in skeletal metabolism and bone composition in fishes [20,73,75,76,77]. Although fatty acids have been one of the most studied nutrients, their effects on bone metabolism have not been well identified [78]. The impact of these nutrients in bone formation varies among the different types of PUFAs [4]. However, different studies have indicated that the consumption of moderate levels of Omega-3 promotes correct skeletal development [79]. Consequently, it is considered that the consumption of these nutrients in adequate amounts can improve bone formation and health [79], while imbalances in dietary levels have been related to several bone disorders and types of skeletal malformations [20,80]. However, the molecular mechanism involved in this beneficial effect remains mostly unknown.

The transcriptomic analysis performed in the larvae of *Argyrosomus regius* fed with experimental diets showed several genes differentially expressed among the different groups. However, among the treatments, the MD showed the most different pattern of expression and the highest number of DEGs. To obtain information about which of these DEGS were related to the control of PUFAs in skeletal development, enrichment analyses employing the GO and KEEG databases were performed. The results indicated that among the terms, the most enriched pathways crucial for the control of skeletal development were obtained. Among them, several were related to membrane transport, cell communication, and signaling control of bone-cell differentiation and activity. Several were also involved in binding and interactions among ECMs and other cellular components, or were related to important cellular processes such as ion transport and calcium metabolism or in the biosynthesis of PUFAs or their subproducts (including pathways related to the antioxidant system and inflammation). The impact on genes involved in these cellular pathways was also considered due to the fact that the onset of several skeletal disorders is related to imbalances in these cellular pathways, leading to increased oxidative stress and high synthesis of pro-inflammatory molecules that negatively impact bone and cartilage metabolism [65,81,82,83,84].

Once the most important enriched pathways related to the control of PUFAs in skeletal development were selected, this study searched for interesting DEGs correlated to them to gain information about the impact of these diets on expression of the individual molecules that control bone formation. However, the knowledge about the function of genes involved in skeletal development in teleosts is still very scarce. Due to this, information obtained in studies with different vertebrates other than fish was employed to unravel the function of the different affected genes in our species. This approach is possible because most skeletal tissues, bone cells, and key factors controlling mammal skeletal development have orthologues in teleosts, where the molecular and cellular mechanisms involved in the regulation of bone morphogenesis seem to be conserved among vertebrates (see reviews [85,86]). Based on this, 96 DEGs were selected among these processes for further study. They were divided into three panels regarding their main effects during bone formation, and due to this, several of these DEGs were included in more than one enriched pathway. The genes were divided regarding if they control the differentiation of bone cells [87,88,89]; they synthesize bone proteins and ECM components [90]; or if they modulate important signaling and cellular pathways that affect bone metabolism [91] such as Ca metabolism and transport [92,93], oxidative stress [94], and inflammation [95].

### 4.1. Molecules That Impact Molecular Pathways Involved in Bone-Cell Differentiation

During the process of bone-cell differentiation, osteoprogenitors acquire specific phenotypes under the control of adequate regulatory factors. The differentiation and activity of terminal chondrocytes, osteoblasts, and osteoclasts in vertebrates are carried out through important transcriptional and signaling pathways. These pathways are controlled by transcription and growth factors, among other signaling molecules produced within the bone-cell lineage and other cell types [51,54,96,97,98,99]. Furthermore, other genes coding for ECM-related proteins were also included in this panel as it has been reported that they contribute to bone-cell differentiation, indicating that mechanisms underlying differentiation are likely more complex than previously appreciated [56,57]. The factors regulating bone-cell differentiation are expressed and control bone-cell differentiation at different points during maturation.

Among the different transcriptional factors impacted by the different dietary treatments, it was observed that the MD down-regulated genes controlling the commitment of mesenchymal cells through osteogenic lineages such as *pparg*, whose inhibition has been related to an increased osteogenic commitment while inhibiting adipogenic commitment [100]. Furthermore, down-regulated genes also promote early differentiation of OB and most CH lineages such as, for instance, *sox10* [101] or *znf219* (which is a transcriptional partner of *Sox9*) [102]. Nevertheless, other crucial factors during the early stages of CH differentiation were up-regulated, such as *barx1* [103], whose attenuation and mutation in zebrafish led to dysmorphic arch cartilage elements due to reductions in chondrocyte condensation [104]. The MD seems to also up-regulate factors involved in more advanced stages of CH differentiation, such as *foxa2* [105,106], that prompt the expression of chondrocyte-proliferating population markers, while decreased transcriptional factors are involved in the terminal phases of CH differentiation and hypertrophy, such as *mecf2c* and *mecf2d* [107] (which furthermore seem to promote OC differentiation [108,109]). The inhibitory effect of OC differentiation can also be appreciated in the suppression of *asxl2*, which is involved in the assembly of transcription factors promoting myeloid differentiation [110]. These results indicate that the MD inhibited genes involved in early commitment and terminal differentiation of the CH lineage while it increased the differentiation of chondroprogenitors to proliferative chondrocytes that synthesize the ECM. Nevertheless, the high level of Omega-3 (HD), significantly increased *sox10*, *znf219*, and *bcl11* (which is a suppressor of premature OB differentiation [111] and is absent in cranial endochondral ossification) [112] in comparison to the MD.

Furthermore, several secreted molecules and growth factors produced within the bone-cell lineage and other cell types regulate the molecular pathways responsible for the control of bone-cell differentiation (see reviews [98,113]). Among the cytokines involved in this function, a few can be highlighted: Hedgehogs, NOTCH, BMPs, TGF-β, and WNTs (see reviews by [114,115,116]), which are indispensable for both direct endochondral and intramembranous ossification [117]. These molecules exert different effects depending on the bone cells and the differentiation stage in which they are expressed. For instance, TGF-β cytokines can exert a dual impact. Their expression presents a pro-osteogenic effect when expressed at the early stages of differentiation by promoting mesenchymal cell commitment into OB or CH lineages and decreasing OC differentiation. However, this cytokine also inhibits terminal differentiation in OB and hypertrophy in CH when expressed in more advanced stages of cell development. Among this family, the expression of *tgfβ2*, which is considered crucial for embryonic skeleton development, was decreased in the MD in comparison to the RD [118]. In addition, among DEGs molecules involved in the Wnt-signaling pathway, which is crucial not only in regulating differentiation of osteoprogenitors into OB and CH but also in promoting the late stage of differentiation of these cells and inhibiting osteoclastogenesis [119,120], were also identified. Among them, expression of *lrp6*, which is a receptor that promotes OB maturation [121] and CH hypertrophy [122], was down-regulated in the MD in comparison to the RD. In addition, *wisp3* expression was increased by both levels of supplementation (MD and HD) and was included because it is able to modulate the Wnt-signaling pathway in different ways. On one hand, their interaction with *lrp6* has been related with an inhibition of the Wnt pathway and the decrease in osteogenesis, while at the same time they inhibit chondrocyte hypertrophy by interacting and inhibiting the effect of IGF-1 [123]. Other genes involved in signaling pathways were impacted by PUFAs levels such as *notch2* and *notch3*, which were down-regulated in the MD in comparison to the HD. Despite these molecules seeming to decrease early OB differentiation by decreasing *runx2* or *Col2* expression [124], their main effect seems to be directed toward increasing OC differentiation in different ways [125]. Among notch ligands, *jag2* was similarly regulated and included in the same cluster, denoting a functional relationship. A gene coding for a ptc receptor *ptch3* was highly up-regulated in the MD, and it was included in the panel because it inhibits the Indian hedgehog pathway (Ihh) delaying chondrocyte differentiation and hypertrophy [126]. In developing cartilage, Ihh is primarily expressed by pre-hypertrophic chondrocytes (immediately prior to hypertrophy) as well as in early hypertrophic chondrocytes. The impact on gene expression exerted by the MD again denoted an inhibitory effect in genes that promote chondrocyte hypertrophy and osteoclastogenesis.

The modulation of several ECM-related proteins was also analyzed because they are involved in cell–cell and cell–matrix interactions with regulatory factors and can modulate cell differentiation. Cell–cell and cell–matrix interactions usually require macromolecular assemblies that are formed by several components that include ECM- and cytoskeletal-associated components, among others. Currently, several studies and reviews have reported on their importance in bone formation, and it is believed that imbalances or loss of these molecules lead to alterations in skeletal development [57,127]. Again, a more pronounced impact on the expression of these molecules was exerted by the MD, as demonstrated via enrichment analysis denoting a high number of genes differentially regulated in pathways related to the ECM, cell communication, and binding. Among the up-regulated genes in the MD in comparison with the RD, mainly molecules that control differentiation by interacting with important molecular regulators of bone-cell development were observed. For instance, the MD positively regulated *mgp* expression, which inhibits OB and CH maturation, and matrix mineralization by binding to BPM2 and blocking their signaling effect. This protein also inhibits osteoclastogenesis regulating the expression of *nfatc1* [128,129]. Contrarily, up-regulated genes in the MD were associated with the promotion of OB and CH differentiation as *cthrc1* (encodes a membrane protein secreted by osteoclasts [130,131]), the transmembranal protein *tmem119* (interacts with BMP to exert an osteogenic effect), or *PKdcc* (a tirosine kinase that phosphorylates a broad range of ECM proteins, including collagens and matrix metalloproteinases, and their disruption seem to delay chondrocyte differentiation [132]). Other pro-osteogenic genes that were up-regulated have a key impact in chondrocyte lineage such as *mustn1* (acting in a transcriptional complex promoting chondrocyte differentiation [133]) and *cldn-18* (a member of the tight junction family of proteins, which also presents an osteo-inductive role due to that fact that it is a negative regulator of RANKL-induced osteoclast differentiation [134]). In addition, genes coding for specific components of cartilage were up-regulated by both levels as the gene coding for a transmembrane proteoglycan (*snorc)*. This gene promotes CH differentiation and cartilage synthesis and its expression in cartilage is highest in proliferating and pre-hypertrophic populations, while it is more restricted in hypertrophic chondrocytes [135].

Furthermore, other proteoglycans, named syndecans, with an important role in endochondral ossification, were identified among DEGs [136]. Syndecans interact with a variety of extracellular matrix components and soluble mediators exerting a critical role in adhesion to the matrix and in modulating matrix deposition [137]. Among them, *sdc*-3 was up-regulated by both levels of Omega-3 in the diet, while *sdc-2* was down-regulated in the MD in comparison to the RD. The different regulations can be related to their functions. Syndecan-3 is related to an osteo-inductive role as it increases during OB differentiation, but its effects have been more intimately associated with the control of chondrocyte proliferation [138]. Syndecan-2 regulates OB and CH differentiation, and its expression is up-regulated in response to osteogenic factors [139]. In a study, the overexpression of *sdc-2* inhibited both osteoblast and osteoclast proliferation, but the higher bone mass observed indicates higher inhibition of osteoclast reabsorption [137]. Furthermore, syndecan-2 exerts a critical role in neo-angiogenesis/angiogenic sprouting during embryonic stages, as proved in different organisms such as zebrafish and mice [137]. Due to this, it is expressed in the final stages of chondrocytes’ differentiation coinciding with the onset of cartilage vascularization during the endochondral process. Furthermore, as in the case of *sds-2*, the expression of genes coding for other molecules involved in cell interactions related to advanced states of endochondral ossification were also down-regulated by the MD in comparison with the RD, such as *sema3a*, which code for a secreted and membrane-associated protein expressed by OC that acts as a coupling factor among bone resorption and formation due to its ability to regulate osteoblasts, chondrocytes, and osteoclasts (reviewed by [140]). It has been associated to pro-osteogenic effects [141] and has also been related to angiogenesis due to being expressed in the pre-hypertrophic and hypertrophic chondrocytes coinciding with the onset of endochondral ossification and vascular invasion [142]. Also, a similar expression pattern was obtained in *Cfm*2, which is also expressed by pre-hypertrophic chondrocytes, regulating actin organization. Mutations of this gene in mice manifested defects of the vertebral column as scoliosis and kyphosis or vertebral fusions [143]. Contrary to the MD, the HD seemed to promote hypertrophy by increasing *mxra8* expression, which is a member of matrix remodeling-associated proteins that regulate the Ihh pathway [144]. Moreover, the different impact of both levels was also observed among genes that have been mostly related to the regulation of OB differentiation such as *itga11b1* (binds with osteolectin and increases osteogenesis by regulating the Wnt pathway) and *macf1*, which is also involved in adhesions and interactions with the cytoskeleton and are positively regulated by osteogenic factors and also promote the expression of bone formation markers (*col1*, *runx2*, *and alp*) and OC differentiation. Also, genes coding for *pkd1* and *cdh2*, which participate in cell–cell adhesion, were down-regulated in the MD in comparison to the RD. *PKd1* enhances OB differentiation, while *cdh2* code for a negative modulator of Wnt signaling [145].

The expression of these molecules indicates that the MD seems to highly impact endochondral ossification due to their special impact in several molecules related to cartilage development, mainly inhibiting molecules participating in pathways that promote CH hypertrophy, but also diminishing OB terminal differentiation and OC proliferation. From these results, we hypnotized that the skeletal development of larvae from the MD group are in a less advanced stage of maturation regarding the structures formed by this process in comparison with the RD and HD groups, which did not present several differences among them. In larvae from the MD group, the cartilaginous plate and cartilaginous matrix seem to be still in formation due to the level of Omega-3-increased transcriptional factors and signaling molecules that promote the population of proliferative chondrocytes but inhibit hypertrophy. The larvae from the HD group, in comparison, showed an increased expression of molecules that promote the hypertrophy of chondrocytes and the proliferation of OB and OC, which are mostly related with a more advanced stage of endochondral ossification when the cartilaginous matrix has been totally formed and starts to be replaced and vascularized. In fact, the HD seems to increase expression of molecules involved in the onset of angiogenesis and tissue vascularization while MD inhibited these genes.

### 4.2. Molecules That Impact the Synthesis of ECM Components

ECM is an assembly of several components, which work together to carry out a wide range of functions. The proteins in the extracellular matrix of bone are often classified into two main groups, structural proteins (mainly collagenous proteins) and proteins with specialized functions (non-collagenous proteins) [56]. Around 180–200 different non-collagenous proteins have been identified and are involved in the regulation of the collagen fibril diameter; serve as signaling molecules, growth factors, or enzymes; or have other functions including proliferation, migration, apoptosis, and differentiation (see review [146]). Some of them have been already cited and included in the panel of genes related to factors controlling bone-cell differentiation. The expression of ECM molecules changes among stages of skeletal formation and bone-cell maturation; due to this fact, some are considered markers that permit identification of different cell populations or stages of skeletal development. A high number of genes related to the synthesis of components of the ECM was identified in the transcriptomic analysis, and this can be related to the increased cell proliferation and enhanced tissue that occurs during the first steps of skeletal formation.

Among the genes coding for collagenous proteins, the MD-up-regulated genes coding for markers of proliferating chondrocytes such as *col2a1a* and *col2a1b* [147] and several other minor components of collagens. Furthermore, this level again seems to decrease expression of genes associated to collagens that impact chondrocyte differentiation promoting hypertrophy and endochondral ossification such as *col6a3* [148] (due to their roles in cell adhesions) or *col13a1* (that interact and bind with regulatory molecules as nidogens or integrins to promote angiogenesis [149]). These collagenous proteins, on the contrary, were up-regulated in the HD in comparison to the MD such as also *col5a3*, which is a regulator of col1 molecules and is more associated to the OB-secreted ECM [150]. The expression of genes coding for other collagenous proteins that are usually associated to stability of base membranes were up-regulated by both levels of Omega-3 supplementation as in the case of those usually present in the base membrane of the cartilaginous matrix such as *col4a4* and *col4a5* (that usually forms a composed helical structure with the protein coded by *col4a6* that was more up-regulated in the HD). This triple helical complex possess anti-angiogenic properties that might be involved in the control of vascularization during cartilage repair and in cartilage homeostasis [151]. Furthermore, genes coding for *col6a2* (inhibits OC differentiation), or *col9a1*, *col9a2*, *and col9a3*, were up-regulated, and they are important components of collagen and exert an anti-angiogenic effect preventing vascularization [152]. However, other components present in the cartilaginous matrix that help their stabilization and participate in endochondral ossification included in this panel were down-regulated by the MD as *nid2* and *cilp2* [153]. Furthermore, the MD decreased the expression of genes coding for pro-angiogenic factors associated also with cartilage hypertrophy such as *angpt2* in comparison to RD, and *angpt1* in comparison to HD. Concretely, *angpt2* code for the most expressed angiogenic factor in cartilage, being more expressed than *angpt1* or *vegf* [154]. Furthermore, the anti-angiogenic effect of the moderate level of supplementation in the MD was confirmed by the impact on *thbs1* and *thbs2*, which encode inhibitors of OB ossification and angiogenesis while promoting OC differentiation [155,156].

The MD treatment up-regulated the expression of cartilage proteoglycans *hapln1*, *acanb* [157], while it down-regulated *kera*, coding for keratan, which is more associated to the OB-secreted matrix [158]. In addition, this level also up-regulated the expression of genes coding for proteins involved in their synthesis such as *csgalnact1* (crucial for aggrecan metabolism) or *chst3a* (that catalyze the sulfation of chondroitin [159]). The positive regulation of Omega-3 PUFAs in gene coding for the main components of the chondrocyte-secreted matrix was also observed in regulation of non-collagenous proteins such as *cnmd1*, which is also considered a major regulator in cartilage development and an inhibitor of angiogenesis [160]. Both levels of supplementation up-regulated genes such as *lama3*, *tnxb* [161], and *matrilin1* [162], which are related to cellular interactions among collagens and other ECM components that increase cartilage synthesis. Finally, a minor but potentially interesting fraction of non-collagenous proteins in the ECM of bone that also exert important roles during endochondral ossification are the enzymes, mainly phosphatases as alkaline phosphatase (ALP) and also matrix metalloproteases such as MMP13. The genes coding for these enzymes were also founded among DEGs. While the gene coding for *alp* (involved in ECM mineralization [163]) is increased significantly in the MD, the expression of *mmp13* (that degrades the ECM) was significantly reduced in the HD in comparison to the RD [164]. Again, it was observed that LC-PUFAs’ dietary level modulates the expression of collagenous and non-collagenous proteins that exert important roles during bone formation. Among the diets, the MD seems to again impact especially during the endochondral process. Concretely, in this panel of genes, the moderate Omega-3 supplementation up-regulated the expression of markers of proliferating chondrocytes and cartilaginous ECMs while inhibiting hypertrophy and angiogenesis. The HD, however, seems to have had a positive impact in molecules that are more related with the OB-secreted matrix or in molecules involved in late stages of endochondral ossification as, for instance, in the genes that promote mineralization, matrix degradation, and vascularization.

### 4.3. Molecules That Impact Crucial Cellular Processes

Molecules that are participating in fatty acid metabolism and synthesis of their lipid mediators, calcium metabolism, antioxidant system, and inflammation were further analyzed. Among the DEGs included in these pathways, the MD seemed to increase all the genes included in this panel that were involved in fatty acid metabolism and binding and have been identified in skeletal cells as fatty acid binding proteins [165] *(fabp2*, *fabp6*), and the *cd36* molecule that is crucial for fatty acid uptake is expressed in OB [166] and OC [167]. Some genes coding for proteins involved in calcium metabolism and mineralization such as *sst* (which is a hypocalcemic factor and seems to inhibit bone and cartilage formation [168]), *anxa2* (that also affects osteogenesis [169]), or s100 proteins such as *s100p* and *s100a10a* (that have been described in mammalian and teleost bone and seem to be crucial for endochondral ossification and seem to supress chondrocyte hypertrophy) were also up-regulated in the MD in comparison with the RD [170]. Contrarily, calcium-dependent proteins involved in cellular signaling coded by *camk2a* and *camk2b1* were down-regulated by the MD. These proteins have been associated with cartilage homeostasis [171]. In fact, expression of *camk2* gene has been related to premature chondrocyte maturation [172].

Several genes involved in the antioxidant system were up-regulated by the MD. Namely, as observed in DEGs, genes coding for glutathione peroxidases (*gpx1*, *gpx4a*, and *gpx8*), which are key suppressors of lipid peroxidation, were expressed in bone cells and are also extremely important in other tissues for cellular detoxification of ROS. The expression of the different GPXs seems to be inhibited by the high level of Omega-3 in the diet, which can indicate a lower protection in these larvae against ROS. This effect was also observed in other fish larval cultures where higher Omega-3 content in the diet reduced their expression [20]. Other antioxidant molecules involved in the reduction of hydrogen peroxides that further modulate osteogenic differentiation like peroxiredoxins (*prdx1*, *prdx2a*, and *prx2b*) [173,174] or *gsto*1 (which is expressed in a wide range of human tissues including bone [175]) were similarly regulated by the MD. Lipid peroxides are common subproducts derived from the metabolism of high energetic PUFAs. These compounds are toxic in higher amounts, and their accumulation results in damage to cellular bio membranes [176]. Furthermore, the accumulation of peroxides due to the insufficient detoxification and defense against them have been related to an increase in osteoclastogenesis, bone resorption, and inhibition of bone formation [81]. These results suggest that larvae fed with the HD might be less protected against oxidative stress, which can be prejudicial for bone health. The lower oxidation exerted by the MD was also evidenced by the lower expression of *pparα*, which is usually highly expressed in tissues presenting higher oxidation [177]. Furthermore, the negative effect of excessive fatty acid oxidation and increased oxidative stress has also been related to positive feedback in inflammation. The inflammatory process is also highly impacted by LC-PUFAs due to their derived lipid mediators (eicosanoids and docosaenoids) usually present with pro- or anti-inflammatory potential. The MD decreased genes involved in the synthesis of lipid mediators promoting inflammatory processes such as *ltb4*, which lead to the release of pro-inflammatory cytokines or genes coding for the synthesis and metabolism of prostaglandins such as *ptgis*, *ptgs2b*, or *ptger2a*. Prostaglandins are important signaling molecules in bone metabolism. PGE2 is synthetized by COX2 and is one of the most biologically active factors in controlling bone metabolism. While lower amounts of PGE2 seem to promote bone formation, their synthesis in high amounts has been related to an increase in inflammation and ROS that seem to be deleterious for bone formation and skeletal health. Furthermore, the MD increased *ephx2*, which converts epoxy fatty acids (EpFAs) in their diols that are not inflammatory. In this panel, we could observe that the MD positively regulated antioxidant defense and calcium transport and inhibited pro-inflammatory genes, while no differences in any of these genes was observed among the HD and RD. In this panel, it was observed that the MD seemed to up-regulate the expression of antioxidants and molecules involved in calcium and fatty acid transport while inhibiting signaling molecules such as inflammatory molecules and calcium-dependent kinases in comparison to the RD.

The results obtained in the transcriptomic analysis are in agreement with authors that have already discovered that Omega-3 PUFAs (EPA and DHA) and their metabolites assist in regulation of cell functions through regulation of gene expression directly, by interacting with transcriptional factors, or indirectly, by influencing the membrane lipid composition and modulating several cell signaling pathways [178]. Since the progression of skeletal development is guided by the expression of different regulators and synthesis of the different components of bone, from the different expressions among the different treatments, we can identify different regulations exerted by the diets in this process and according to the stage of skeletal development. Analyzing the results on gene expression, the MD presented a very different genetic profile in comparison to the other treatments. This different pattern of expression seems to be related to a different stage of skeletal development. Mainly, the differences among diets seem to impact a high number of genes involved in the differentiation of bone cells and in the synthesis of the ECM. Concretely, a high regulation of chondrocyte-related genes and cartilage ECM components, among other molecules involved in endochondral ossification, has been observed, which indicates that differences among diets mainly affect the structures formed through this mechanism. The MD up-regulated biomarkers were mostly related to proliferating chondrocytes, while inhibiting genes that were related to a more advanced stage of chondrocyte differentiation such as hypertrophy. Among these genes, we can include several transcription factors and also proteoglycans or other components of cartilage ECM. Furthermore, this diet seems to also inhibit a high number of genes that are related to the control of OB and OC differentiation. These genes, however, seem to be more highly expressed in the HD and did not present several differences in the RD. Therefore, while the intermediate level seems to inhibit the genes promoting the final steps of endochondral ossification, the HD seems to up-regulate the genes promoting these processes; for instance, it increased genes that promote chondrocyte hypertrophy and ECM mineralization such as angiogenic molecules and genes related to OB and OC proliferation required for the replacement of the ECM formed by the cartilage. Due to this, these genes reflected an advanced state of endochondral ossification.

These results are in agreement with the analyses of the skeleton performed in larvae during the trial [179].These analyses, performed at the end of the trial at 42 DAH, indicated that dietary supplementation with the highest level of Omega-3 (HD) improved larval growth performance and accelerated the process of skeletal development and the rate of skeletal mineralization. However, at 28 DAH, larval growth and skeletal development were similar among the HD and RD, while the larva from the MD was slightly smaller and presented less developed skeletons. Regarding the frequency of skeletal anomalies, the moderate level of Omega-3 supplementation in the MD decreased the frequency of the malformed larvae observed during the trial in comparison to the RD that did not present significant differences with to HD. Concretely, the MD exerted a reduction in the frequency of anomalies presented in structures that are formed via endochondral ossification as vertebral arches and spines. This beneficial effect observed during the trial in larvae fed with the MD could be associated, hence, with better regulation of the molecular and cellular mechanisms through which fatty acids are able to control skeletal development and the concrete endochondral ossification process.

The special impact of Omega-3 supplementation in cartilage metabolism that has been observed in the transcriptomic analysis has also been identified by several authors during in vivo and in vitro experiments. Koren et al., 2014 [180] indicated that treatment of chondrocytes with Omega-3 FAs increased chondrocyte proliferation, proteoglycan synthesis, and the expression of collagen type II. These results demonstrate the beneficial effect of n-3 LC-PUFAs’ increased chondrocyte proliferation and differentiation in vitro. These promising effects of Omega-3 supplementation have been also identified in reared fishes. Several studies have already identified that Omega-3 dietary supplementation promote larval growth and bone formation and increase the rate of bone mineralization while, to the contrary, deficiencies in n-3 LC-PUFAs, as demonstrated with DHA, produced low mineralization and down-regulated the expression of osteogenic markers such as ALP in gilthead seabream (*Sparus aurata*) larvae [181]. These results were in agreement with others obtained from *Sparus aurata* larvae, where the deficiency of DHA has also been associated with the lowest number of mineralized vertebrae and with lower mineral content [19].

However, other authors have identified a negative impact on skeletal development with Omega-3 dietary supplementation. The deleterious effects have usually been observed with an excess of these fatty acids. For instance, an excess of EPA in Senagalese sole (*Solea senegalensis*), rather than enhancing growth and development, had a detrimental effect on morphogenesis [182]. Along these lines, the excess of DHA seems to lead to an increase of bone deformities in the presence of supernumerary skeletal structures, as has been recently observed in piperkech larvae (*Sander lucioperca*) [20]. Similarly, the elevation of both n-3 PUFAs, has also been demonstrated to be detrimental in European seabass (*Dicentrarchus labrax*) and induced both cephalic and vertebral column deformities, while a moderate level of PUFA with diets containing 1.1% to 2.3% EPA and DHA, decreased this incidence [183]. Similar results were also observed in studies on fast-growing species as in long fin yellowtail *(Seriola rivoliana)* larvae; the studies have demonstrated that the incidences of some anomalies such as kyphosis and lordosis increased along with the dietary DHA contents [184]. These deleterious results have been mainly related to the impact of excessive levels of Omega-3s in the important cellular pathways cited above. For instance, several authors associated the increased occurrence of bone deformities, with oxidative damage, as a consequence of an excess of n-3 LC-PUFAs. In fact, deleterious effects were observed when high Omega-3 levels were included in diets accompanied by a low level of antioxidants in larval stages of other species such as in sea bass [185] or *sparus aurata* [19]. This is due to the fact that cartilage and bone cells or matrix vesicles are composed of PUFAs that are very prone to suffer lipid peroxidation, and this can represent a risk for bone health [73,186,187]. These results could be in agreement with our results, due to the diet with a moderate level of Omega-3 seeming to present a higher capacity to combat oxidative stress due to the higher expression of antioxidant molecules and reduced expression of molecules that promote fatty acid oxidation. Furthermore, other authors have also related the negative effects of excessive Omega-3 levels with a deregulation in the synthesis of inflammatory mediators. The beneficial effect of the MD in skeletal development can be also related to the better regulation carried out by this diet in molecules involved in that function, due to the fact that a decreased expression in the larvae fed with this diet can be observed in the molecules that participate in the synthesis of pro-inflammatory lipid mediators. Furthermore, the beneficial regulation of cellular pathways carried out by the MD can be also evidenced in the up-regulated expression of molecules involved in calcium transport due to that fact that it is believed that Omega-3 LC-PUFAs could also benefit skeletal health, enhancing the uptake of Ca. The different modulations exerted in the genetic expression profiles by both levels of PUFAs can be associated with the impact of these nutrients on the correct development of the skeleton during larval stages. While moderate Omega-3 levels seem to delay endochondral ossification, an increase in the synthesis of cartilage, and inhibition of OB proliferation, the high level seems to increase the skeletal formation rate by enhancing OB differentiation. Furthermore, the moderate level also impacted cellular pathways differently, increasing expression of antioxidant molecules and diminishing the expression of molecules related to inflammatory lipid mediators. The different regulation among all these pathways can be related to the differences in growth, skeletal formation, and incidences of skeletal anomalies at the end of the trial. However, further studies are required to investigate if the different modulations observed among of the genes that we selected can be effectively associated to the onset of anomalies.

## 5. Conclusions

Omega-3 PUFAs are able to regulate the expression of important genes involved in skeletal formation. Therefore, exploring the modulatory impacts exerted by different formulations on the expressions of molecules involved in the control of skeletal formation provided new insights about how these nutritional factors are able to modulate this process. Employing the available information about the most important molecular and cellular pathways affected by dietary LC-PUFAs that are involved in the control of bone metabolism, we have selected candidate genes that can be relevant in the onset of skeletal anomalies. The results demonstrated that levels of dietary Omega-3 PUFAs are able to impact the expression of several molecules involved in pathways that control bone-cell differentiation, the synthesis of ECM components and cellular pathways such as calcium metabolism, oxidative stress, or the synthesis of inflammatory mediators that also impact bone composition and health. Among these molecules, Omega-3 inclusion seems to especially impact molecules that are involved in cartilage metabolism related to endochondral ossification. However, a different regulation of these molecules was observed in both levels of supplementation at 2.6% and 3.6% Omega-3 in the diet. Thus, more studies are necessary to confirm whether the different regulations exerted in these promising genes are effectively related to the beneficial effects associated to dietary supplementation with Omega-3 and to ascertain the imbalance with which these molecules can be related regarding the onset of skeletal anomalies.

## Figures and Tables

**Figure 1 biomolecules-14-00056-f001:**
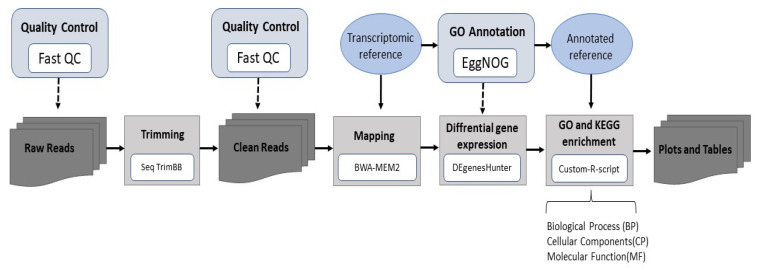
RNAseq analysis pipeline. Data processing was performed with the computational resources of the Andalusian Bioinformatics Platform located at the University of Málaga (Spain).

**Figure 2 biomolecules-14-00056-f002:**
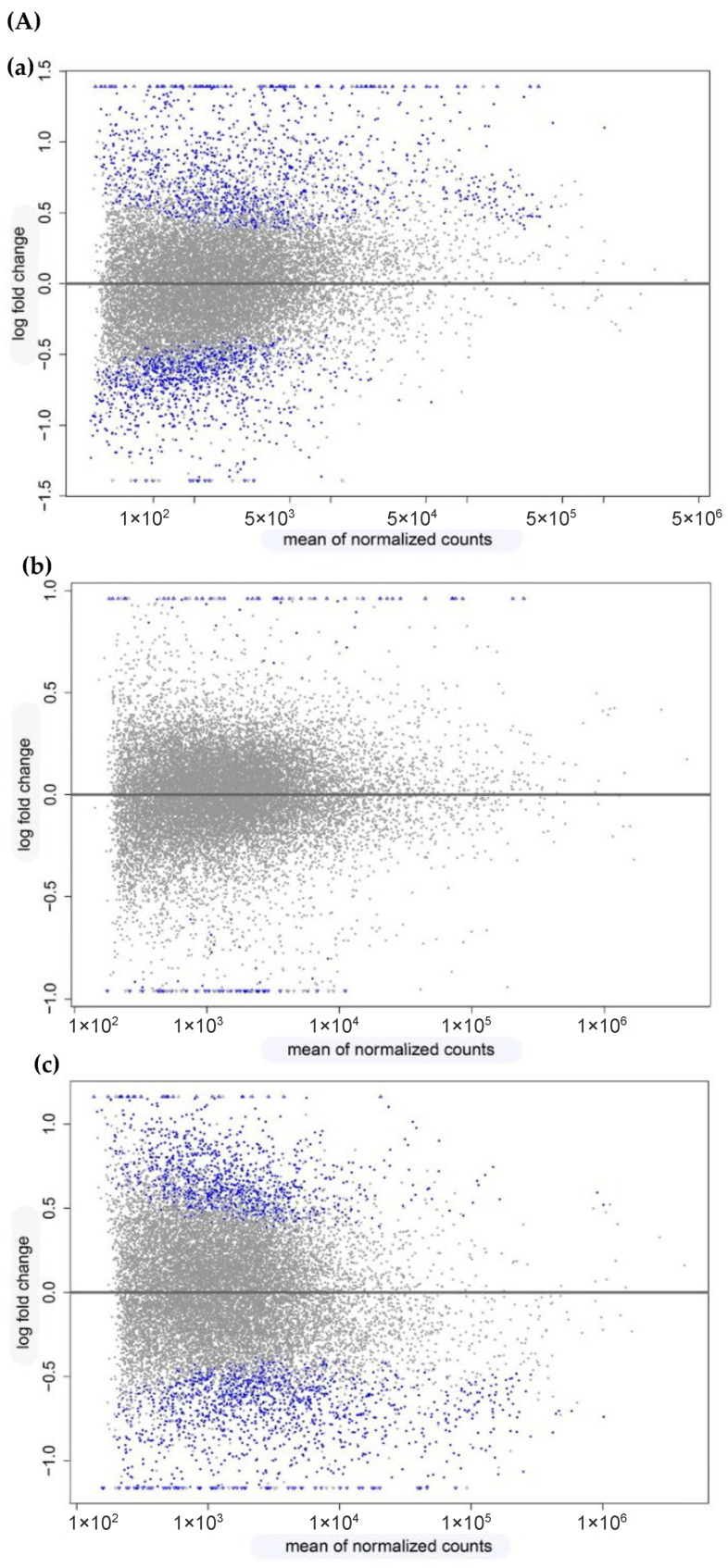
Visualization of gene expression datasets in *Argyrosomus regius* larvae fed with different level of Omega-3 (commercial, medium, and high) at 28 DAH. (**A**) These figures show the plots of log2 fold-change against mean of normalized counts (MA plot) of all transcriptome genes from pairwise comparisons: (**a**) MD vs. RD, (**b**) HD vs. RD, and (**c**) HD vs. MD. Blue dots represent genes that passed the threshold FC > 1.5 (*p*.adj < 0.05, false discovery rate correction, FDR). Grey dots represent expressed genes that did not pass the thresholds in order to be considered true DEGs. MD, Medium Diet; HD, High Diet; RD, Reference Diet. (**B**) Venn diagram of the number of genes declared as true DE for each comparison. The figure shows shared and unique DEGs obtained in the pairwise comparisons between treatments. The number of true DE genes that were obtained in MD vs. RD is surrounded in orange. The number of true DE genes that were obtained in HD vs. RD is surrounded in green, and the number of true DE genes that were obtained in HD vs. MD is surrounded in blue.

**Figure 3 biomolecules-14-00056-f003:**
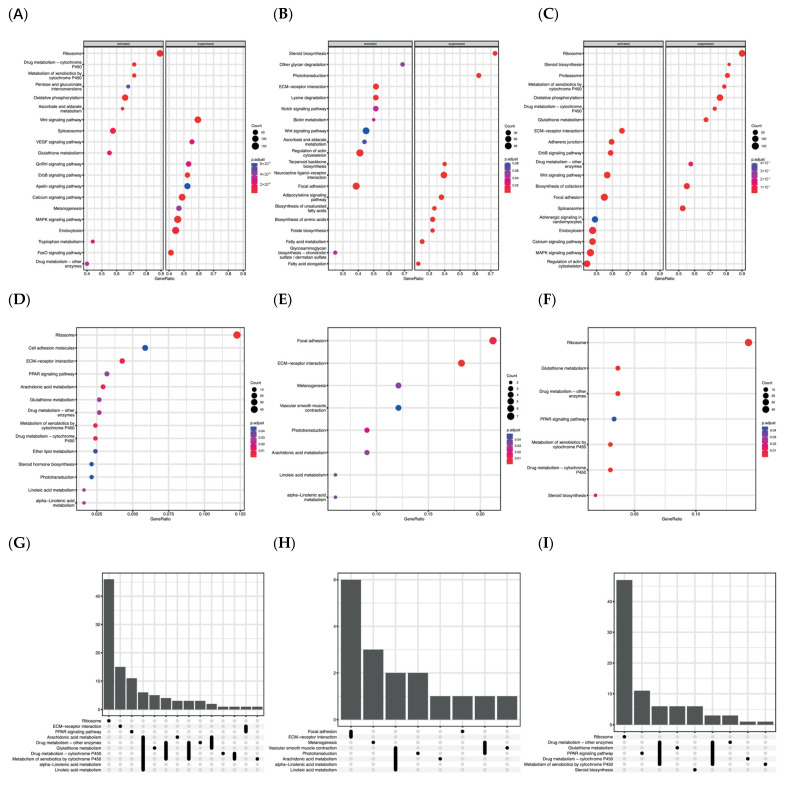
Kyoto Encyclopedia of Genes and Genomes (KEGG). KEGG pathway functional analysis of differentially expressed genes during skeletal development of *Argyrosomus regius* fed with different Omega-3 dietary levels at 28 DAH. (**A**–**C**) represent facet dot-plots of nominally significant up-regulated and down-regulated KEGG pathways enriched among all expressed genes in the different comparisons MD vs. RC, HD vs. RD, and HD vs. MD according to GSEA results. (**D**–**F**) dot-plots represent the most enriched KEGG pathways among all differently expressed genes in the different comparisons MD vs. RC, HD vs. RD, and HD vs. MD, according to ORA results. In all these cited figures, the x-axis shows the ratio of differentially expressed genes in each term relative to the total number of genes in that term. The y-axis represents the term associated to each enriched pathway. The size of the dot indicates the number of differentially expressed genes in that term, and the color of the dot shows the pathway enrichment significance according to *p*-value. (**G**–**I**) Upset-plots presented the number of common elements among GO terms in our functional enrichment analysis for each comparison of MD vs. RD, HD vs. RD, and HD vs. MD, respectively. The vertical bars indicate the common elements in the sets, indicated with dots under each bar. The single points represent the number of unique elements in each group.

**Figure 4 biomolecules-14-00056-f004:**
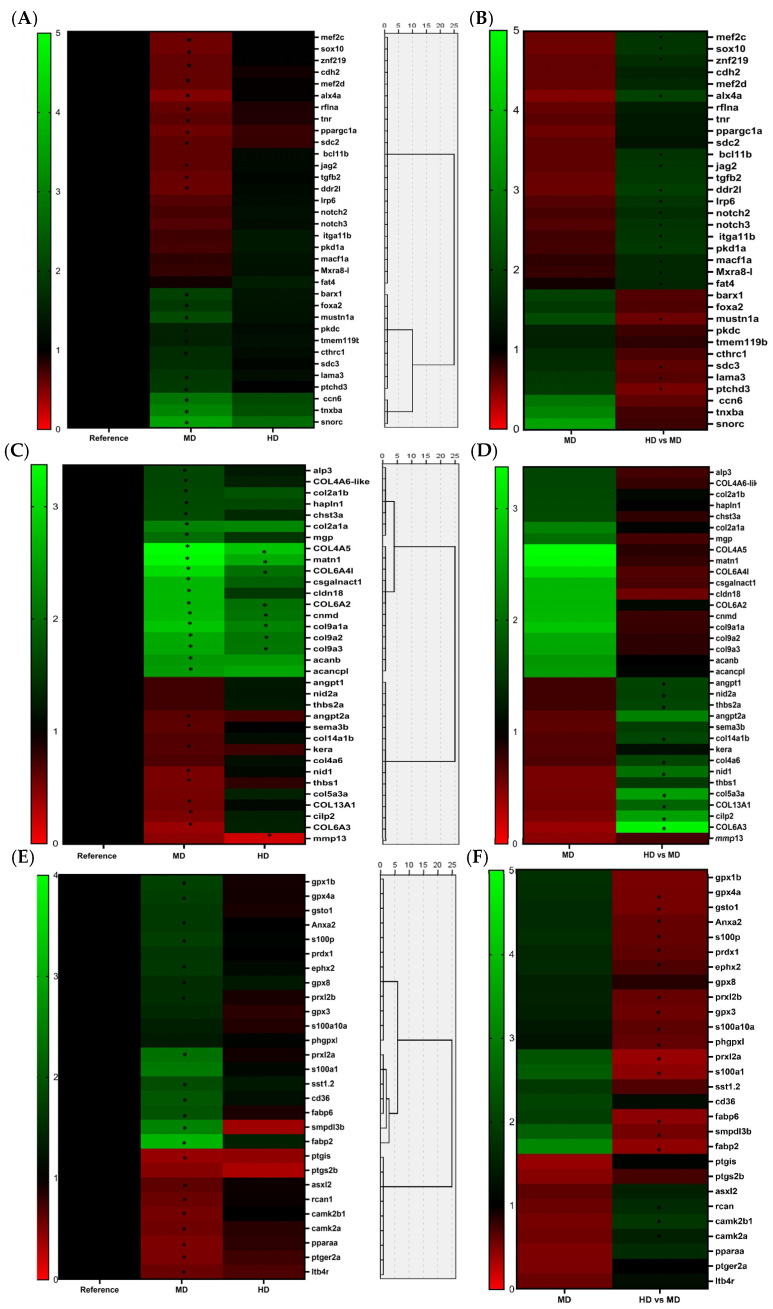
Heatmaps and dendrograms showing the hierarchical clustering of DEGs obtained employing Euclidean distances and the Ward method. Heat maps of DEGs obtained from the comparison between experimental groups treated with different levels of Omega-3 supplementation were generated with Graphpad. Expressions of genes in MD vs. RD and HD vs. RD are showed in graphics (**A**,**C**,**E**), while expressions in HD vs. MD are shown in separate graphics (**B**,**D**,**F**). Red bands represent low levels of gene expression; green bands represent high levels of gene expression. Due to the fact that some genes were not considered as DEGs in all the comparisons due to not passing the double threshold of fold-change and FDR value, the asterisks indicate the comparison for which these genes were considered true DEGs. The first panel includes genes involved in skeletal cell differentiation; the second panel includes genes coding for molecules involved in the synthesis of ECM components, binding among cells, and cells with the extracellular matrix; and the last panel includes genes involved in important signaling cellular pathways. Panel (**A**): myocyte-specific enhancer factor 2C (*mef2c*); SRY-box transcription factor 10 (*sox10*); zinc finger protein 219 (*znf219*); cadherin-2 (*cdh2*); myocyte enhancer factor 2d (*mef2d*); ALX homeobox 4a (*alx4a*); refilinA (*rflna*); tenascin R (*tnr*); peroxisome proliferator-activated receptor gamma coactivator 1 alpha (ppargc1a); syndecan-2 (*sdc2*); bcl11 transcription factor B b (*bcl11b*); protein jagged-2 (*jag2*); transforming growth factor beta 2 (*tgfβ2*); discoidin domain receptor family, member 2 like (*ddr2l*); low-density lipoprotein receptor-related protein 6 (*lrp6*); notch receptor 2 (*notch2*); notch receptor 3 (*notch3*); integrin alpha 11b (*itga11b*); polycystic kidney disease 1a (*pkd1a*); microtubule actin crosslinking factor 1a (*macf1a*); matrix remodelling-associated protein 8-like (*mxra8-l*); FAT atypical cadherin 4 (*fat4*); BARX homeobox 1 (*barx1*); forkhead box A2 (*foxa2*); musculoskeletal embryonic nuclear protein 1a (*mustn1a*); protein kinase-like domain containing (*pkdc*); transmembrane protein 119b (*tmem119b*); collagen triple helix repeat containing 1 (*cthrc1*); syndecan-3 (*sdc3*); laminin, alpha 3 (*lama3*); patched domain containing 3 (*ptchd3*); cellular communication network factor 6 (*ccn6*); tenascin Xba (*tnxba*); secondary ossification center-associated regulator of chondrocyte maturation (*snorc*). Panel (**B**): alkaline phosphatase 3 (*alp3*); collagen alpha-4 (VI) chain-like (*col6a4-like*); collagen, type II, alpha 1b (*col2a1b*); hyaluronan and proteoglycan link protein 1 (*hapln1*); carbohydrate (chondroitin 6) sulfotransferase 3a (*chst3a*), collagen, type II, alpha 1a (*col2a1a)*, matrix Gla protein (mgp); collagen alpha-5 (IV) chain (*col4a5*); matrilin 1 (*matn1)*; collagen alpha-4 (IV) chain (*col4a4*); chondroitin sulfate N-acetylgalactosaminyltransferase 1 (*csgalnact1*); claudin 18 (*cldn18*); collagen alpha-2 (VI) chain (*col6a2*); chondromodulin (*cnmd*); collagen, type IX, alpha 1a (*col9a1a*); collagen, type IX, alpha 2 (*col9a2*); collagen, type IX, alpha 3 (*col9a3*); aggrecan b (*acanb*); aggrecan core protein-like (*acancpl*); angiopoietin 1 (*angpt1*); nidogen 2a (*nid2a*); thrombospondin 2a (*thbs2a*); angiopoietin 2a (*angpt2a*); semaphoring 3B (*sema3b*); collagen, type XIV, alpha 1b (*col14a1b*); keratocan (*kera*); collagen, type IV, alpha 6 (*col4a6*); nidogen-1 (*nid1*); thrombospondin-1 (*thbs1*); collagen, type V, alpha 3a (*col5a3a*); collagen alpha-1 (XIII) chain (*col13a1*); cartilage intermediate layer protein 2 (*cilp2*); collagen alpha-3 (VI) chain (*col6a3*); collagenase 3 (*mmp13*). Panel (**C**): glutathione peroxidase 1b (*gpx1b*), glutathione peroxidase 4a (*gpx4a*), glutathione S-transferase Omega-1 (*gsto1*); annexin A2 (*Anxa2*); protein S100-P (*s100p*); peroxiredoxin 1 (*prdx1*); epoxide hydrolase 2 (*ephx2*); glutathione peroxidase 8 (*gpx8*); peroxiredoxin like 2B (*prxl2b*); glutathione peroxidase 3 (gpx3); S100 calcium-binding protein A10a (*s100a10a*); phospholipid hydroperoxide glutathione peroxidase-like (*phgpxl*); peroxiredoxin-like 2A (*prxl2a*); S100 calcium-binding protein A1 (*s100a1*); somatostatin 1, tandem duplicate 2 (*sst1.2*); CD36 molecule (cd36); fatty acid-binding protein 6 (*fabp6*); sphingomyelin phosphodiesterase acid-like 3B (*smpdl3b*); fatty acid-binding protein 2 (*fabp2*); prostaglandin I2 (prostacyclin) synthase (*ptgis*); prostaglandin-endoperoxide synthase 2b (*ptgs2b*); asxl transcriptional regulator 2 (*asxl2*); regulator of calcineurin 1 (*rcan1*), calcium/calmodulin-dependent protein kinase (CaM kinase) II beta 1 (*camk2b1*); calcium/calmodulin-dependent protein kinase II alpha (*camk2a*); peroxisome proliferator-activated receptor alpha a (*pparaa*); prostaglandin E receptor 2a (*ptger2a*); leukotriene B4 receptor (*ltb4r*).

**Table 1 biomolecules-14-00056-t001:** Primer’s design for *Argyrosomus regius*. Primers were employed for validation of genetic expression.

Gene	Primer Pair	Primer Sequence (5′-3′)	Length (bp)	GenBank Reference or Publication
*sp7*	Sp7_fwd	TTCCTTTTGCGGCTTCAGAG	142 bp	GFVG01056384.1
	Sp7_rev	GCCTGCACACACACATACAA		
*mgp*	mgp_fwd	GCTGGCATTGAATCCCACAT	171 bp	GFVG01044233.1
	mgp_rev	TGTTTCGGTCACCATCCACT		
*bmp2*	bmp2_fwd	TGTGGAATTTATCGGAGCCCA	113 bp	GFVG01014899.1
	bmp2 _rev	CGAGCAGCAGTACCATGAGA		
*ef1a*	*ef1a* _ fwd	GGTGCTGGACAAACTGAAGG	161 bp	Ruiz et al. (2019) [42]
	*ef1a* _ rev	GAACTCACCAACACCAGCAG		
*tub*	tub _ fwd	GGAGTACCCCGATCGTATCA	196 bp	Ruiz et al. (2019) [42]
	tub _ rev	AGATGTCATACAGGGCCTCG		

**Table 2 biomolecules-14-00056-t002:** Summary of RNA sequencing, assembly, and mapping of larvae of *Argyrosomus regius* at 28DAH.

	Reference Diet			Medium Diet			High Diet		
	R1	R2	R3	M1	M2	M3	H1	H2	H3
**Total HQ reads**	180,488,089	164,478,326	163,786,858	152,078,855	179,103,633	253,267,203	164,384,967	150,515,239	163,786,580
**Mapped reads**	118,135,719	111,751,235	114,019,947	105,130,555	118,434,455	169,468,930	111,221,544	105,927,381	114,019,947
**Mapping rate (%)**	64.45%	67.94%	69.61%	69.13%	66.13%	66.91%	67.66%	70.38%	69.61%

## Data Availability

The data presented in this study is available in the current manuscript, raw data or further information is available on request from the corresponding author.

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
