# Peer review of "Exploring Omega-3′s Impact on the Expression of Bone-Related Genes in Meagre (*Argyrosomus regius*)"

_biomolecules, 2023, doi:10.3390/biom14010056_

Round 1

Reviewer 1 Report

Comments and Suggestions for Authors

The article is written very well and clear, it is interesting and discussed on the current and important topic of creating food compositions that will maintain and improve the health of fish, and also have an important applied result - reducing the number of fish with anomalies of skeletal development. The methodology of the article is also excellent, using modern research methods.

1. What is DAH in the Abstract?

2. Introduction. Please provide two-three sentences about the studied fish and it is importance in human consumption since the fish is under commercial aquaculture. It is possible to provide one sentence about the aquculture of this fish worldwide or in certain countries.

3. Line 1196: LC-PUFA... 

4. I do not see supplementary matherial to check the composition of diets. Please provide. 

5. Please check the punctuation (spaces between dots, commas etc.) throught the text. 

Reviewer 2 Report

Comments and Suggestions for Authors

I found this manuscript interesting and based on a well-conducted study which led to important results for the investigated species.

Please take care to double-check pagination following the journal requirements (highlighted parts, enumeration of subchapter of the discussion section, etc.) and especially for the references which are not well-referenced. Double-check all the scientific names to be sure of italics style. Try to improve, as much as you can, the quality of Figures 1 and 3.

Regarding the functional enrichment analyses, please introduce better and contextualize the use of Larimichthys crocea as a reference genome in this study. 

Moreover, a more in-depth introduction and discussion about the genes shown in Table 1 should provided to better contextualize your results nd the research question/conceptualization of this study.

Best regards

The Reviewer

Reviewer 3 Report

Comments and Suggestions for Authors

The MS entitled “Exploring Omega-3's Impact on the expression of Bone-Related Genes in Meagre (Argyrosomus regius)” described an interesting research topic. These results offer important insights into the impact of dietary Omega-3 PUFA on genes involved in the main molecular mechanism and cellular processes controlling skeletal development in larval fish. Here are some comments:

1.     Line 74don’t need to use capital letter for the initials of Eicosapentaenoic acid (EPA), Docosahexaenoic acid (DHA) and Arachidonic acid (ARA)

2.     Is DHA+EPA directly added into the reference commercial feed? Is there a control diet without DHA and EPA supplementation? What is the ratio of DHA:EPA? Please clarify these issues in the M&M section. Supplementary Table 1 and 2 include key information. Please move the into the main text, instead of putting them as Supplementary Tables.

3.     For RNA seq, the sampling size, i.e., six fish, is really too small. Why do you use so few samples?

4.     Why do you set the [fold change] > 1.5? Please provided a supporting material

5.     Table 2, please use MS word format table, instead of a picuture

6.     Figure 3, the figures are too blurred to read. Please provide higher-quality figures.

7.     The Results part is too detailed. Please make it concise.

8.     The discussion part is generally accepted. But it also needs to be more concise too.  

Round 2

Reviewer 3 Report

Comments and Suggestions for Authors

The authors have addressed my concerns. No other comment.